# A Proposal for a Multidisciplinary Integrated Oral Health Network for Patients Undergoing Major Orthopaedic Surgery (IOHN-OS)

**DOI:** 10.3390/geriatrics9020039

**Published:** 2024-03-19

**Authors:** Matteo Briguglio, Thomas W. Wainwright, Marialetizia Latella, Aurora Ninfa, Claudio Cordani, Cecilia Colombo, Giuseppe Banfi, Luca Francetti, Stefano Corbella

**Affiliations:** 1Laboratory of Nutritional Sciences, IRCCS Orthopedic Institute Galeazzi, 20161 Milan, Italy; 2Orthopaedic Research Institute, Bournemouth University, Bournemouth BH8 8FT, UK; 3Physiotherapy Department, University Hospitals Dorset NHS Foundation Trust, Bournemouth BH7 7DW, UK; 4Operational Unit of Phoniatry, ASST Fatebenefratelli-Sacco, 20154 Milan, Italy; 5Department of Biomedical, Surgical, and Dental Sciences, Università degli Studi di Milano, 20122 Milan, Italy; 6Scientific Direction, IRCCS Orthopedic Institute Galeazzi, 20161 Milan, Italy; 7Orthopaedic Biotechnology Laboratory, IRCCS Orthopedic Institute Galeazzi, 20161 Milan, Italy; 8Faculty of Medicine and Surgery, Vita-Salute San Raffaele University, 20132 Milan, Italy; 9Operational Unit of Odontostomatology, IRCCS Orthopedic Institute Galeazzi, 20161 Milan, Italy

**Keywords:** geriatric dentistry, orthopedics, hospital to home transition, oral hygiene, patient care management, malnourishment, multidisciplinary health teams, preventive dentistry, dietitian, enhanced recovery after surgery

## Abstract

The passing of the years of life physiologically leads to the accumulation of changes in tissues in the oral cavity, influencing dentition, chewing and swallowing mechanisms, and the oral microbiota. Some diseases and medications can aggravate oral symptoms and negatively influence eating behaviours, increasing the likelihood of becoming malnourished. This could make older individuals more vulnerable to complications when undergoing major orthopaedic surgery. Hidden infection foci in the oral cavity are a recognised cause of post-operative periprosthetic joint infections. Dysfunctional oral problems might also compromise feeding after surgery when good nutrition represents a fundamental aspect of a proper recovery. To manage these shortcomings, in this article, the authors hypothesise a multidisciplinary path of care named the Integrated Oral Health Network applied to major Orthopaedic Surgery (IOHN-OS). This peri-operative initiative would include pre-operative oral health screening and risk management by a dental team, patient education programmes before and after surgery, and bedside gerodontology actions like oral care and meal and eating support for fragile individuals. The IOHN-OS has the potential to reshape the concept of suitability for major orthopaedic surgery and generate momentum for designing community-based surveillance programmes that can keep the mouths of older subjects healthy for a long time.

## 1. Introduction

Surgery is the reference approach when there is chronic hip, knee, or back pain that reduces a patient’s functional independence and quality of life. The age-related deterioration of joints and bones is the fundamental cause, and, therefore, the majority of patients are older adults. The insertion of joint endoprostheses or spine stabilisers leads to good results in most patients. However, a number of cases do not properly recover, indirectly producing far-reaching human and economic consequences [1]. The common denominators in these cases are smoking, polymorbidity, and malnutrition [2]. Improving outcomes requires innovations, both in surgery and in the peri-operative path of care [3]. This approach is more likely to correct modifiable physical, mental, and social conditions of vulnerability [4]. This notion is not new, and guidance stating that a holistic approach can counteract the reduction in the intrinsic capacity of older individuals has already been issued by the World Health Organization [5]. In orthopaedic surgery, examples of multidisciplinary and multimodal approaches are the Enhanced Recovery After Surgery (ERAS) [6], which has previously been applied in hospitals for some time, bringing notable improvements to clinical practice, and the Healthy Eating, Physical Activity, and Sleep hygiene (HEPAS) [7], which has remained a conceptual proposal for a novel territorial organisational model. Both initiatives demand alliances between orthopaedic surgeons, anaesthetists, case-manager nurses, physiotherapists, orthopaedic technicians, dietitians, and psychologists. A continuum of this collaborative care during home transitions and in the community might have the desired effect of guaranteeing the maintenance of a patient’s well-being in the long term [8]. To further enhance the quality of care of multidisciplinary approaches in orthopaedic surgery, we theorise a new in-hospital path that would integrate a dental team into the professional community that takes care of the older orthopaedic patient.

## 2. Ageing Poses a Risk to the Oral Cavity of Older Individuals

Tooth loss is not inevitable. However, ageing has a negative impact on teeth structure, salivation, chewing and swallowing mechanisms (i.e., presbyphagia) [9], oral microbiome [10], and oropharyngeal sensitivity. A decrease in gustatory and olfactory functioning influences the hedonic appreciation of food and can push a person towards increasingly rewarding food choices [11]. Food predispositions mainly drive the subject’s choices towards the more cariogenic or less healthy food nevertheless (e.g., sweeter and saltier) [12]. Xerostomia is known to impair cephalic digestion [13]. It is not uncommon for patients with poor oral moisture to modify their eating behaviours to avoid dry foods [14]. Low salivation increases the risk of oral diseases due to the lack of an optimal concentration of species-specific antimicrobial peptides, thus allowing tooth decay-causing bacteria to grow freely [13]. A recession of gingival tissues exposes the root surface to this contaminated environment. A lack of enamel and progressive demineralization make the area more susceptible to cavities and caries. Bone resorption and negative calcium balance in osteoporotic patients also affect trabecular bone parameters like number, thickness, and connectivity [15], further increasing alveolar porosity [16]. Alveolar bone, maintained by healthy oral conditions and reinforced by biting occlusal forces [17], deteriorates and no longer supports the teeth. Edentulism and ill-fitting dentures are associated with worse bite force and chewing efficiency [18]. Patients’ food choices will be further diverted to foods with a soft texture [19].

## 3. Impact of Medications and Illnesses on the Oral Health of Older Patients

Oral problems can arise due to concomitant illnesses prevalent in the older population or can worsen after the chronic use of drugs that treat the same diseases. Increased susceptibility to oral infections is a consequence of xerostomia-derived diabetes mellitus [20], with hyposalivation also being initiated by some antihypertensive medications and tricyclic antidepressants [21]. Glossitis, caries, and maxillary and mandibular osteoporosis are all common manifestations of thyroid dysfunction [22]. Swallowing disorders occur in advanced Parkinson’s disease and other neurodegenerative conditions involving hypotonia of the oral muscles [23]. Similar disorders are experienced in the case of severe gastropathy and gastroesophageal reflux. On the other hand, the use of aluminium-containing antiacids to relieve gastric hyperacidity can significantly increase the urinary and faecal excretion of calcium if not balanced by an adequate dietary intake, as well as phosphorus [24]. Similar consequences on calcium homeostasis and that of other minerals essential for dental health (e.g., zinc, magnesium, potassium) are elicited by loop diuretics [25], which are frequently used in disorders of sodium balance. The bone side effects of drugs include sparse facial bone structures and can also lead to consequences that are rare, serious, and definitive, such as the osteonecrosis of the jaw derived from prolonged therapy with antiresorptive bisphosphonates [26].

## 4. The Vicious Cycle between Oral Health and Nutrition in Older Adults

Oral health problems can be detrimental to an individual’s nutritional status [14,19]. Food choices are guided by the degree of masticatory and swallowing ability [19]. Hence, older adults with dentures may not be able to easily eat raw vegetables and nuts [27]. Softer and easier-to-chew foods are usually preferred, but they tend to have less nutritional value [28]. The prolonged consumption of texture-modified food groups can be harmful, analogous to extended cooking to soften the texture. In fact, this practice has been plainly recognised to deplete the content of heat-labile vitamins [29]. The mouth also mirrors the subject’s degree of commitment to oral hygiene. Individuals with cognitive dysfunction have generally poorer oral health [30], which in turn can indicate how physically inactive [31] or physically dependent [32] the subject is. Individuals who are already malnourished may show typical oral signs. Among the first consequences are effects on the mucosa due to its rapid turnover. Glossitis and angular cheilitis can occur in the case of inadequate levels of B-complex vitamins or iron, whose deficiency anaemia is associated with gingivitis, oral ulcers, glossitis, and dental decay [33]. L-ascorbic acid deficiency causes periodontal and gum diseases and bleeding on probing [34]. In the case of heartburn, patients often limit their intake of citrus fruits, further increasing the risk of scurvy. Folate and B12 deficiencies, present in approximately 1 in 15 older adults [35], are associated with a fiery red sore tongue and ulcer formation. In addition to deficiencies of water-soluble vitamins, older adults who use bile acid sequestrants like cholestyramine are also exposed to reduced absorption of lipophilic vitamins, all of which play an important role in oral health maintenance [36].

## 5. Unhealthy Oral Health Conditions Might Pose a Risk to Recovery after Surgery

It is not known what causes an impact first, whether it is the ageing process, malnutrition, or the oral consequences of medications and diseases. It is likely that these factors are intertwined in a complex, vicious cycle. It is estimated that there are around 100 million people over the age of 65 living on this planet with a disabling musculoskeletal condition who would benefit from surgery [8]. Among older adults scheduled for joint replacement surgery, only 1 in 10 eat healthily [37], 3 in 10 show systemic signs of micronutrient deficiencies [38], and 4 in 10 are malnourished before surgery [39]. Given the malnutritional phenotype of the older patient undergoing major orthopaedic surgery, it is not surprising that oral problems in this cluster have a higher prevalence than their healthy counterparts. Periodontal diseases and the decay of permanent teeth affect one in five older people globally, with a similar prevalence of edentulism, which grows exponentially after the age of 65 [40]. Conversely, oral problems that hide an active focus of infection afflict as many as one in three older patients waiting for a hip or knee replacement [41]. This is particularly relevant to pre-operative risk stratification. A healthy oral cavity harbours an ecological balance that depends on the conditions of the human host and living microorganisms. A disruption of this balance due to the worsening of the hosts’ strength of constitution or changes in the oral environment can lead to bacterial overgrowth, aspiration pneumonia, and translocation to other tissues [42]. It has been a common belief for more than thirty years now that infection foci in the oral cavity at the time of major orthopaedic surgery may increase the risk of periprosthetic joint infection (PJI) [43]. Intermittent/persistent bacteraemia from dental infections is known to be a source of haematogenous seeding, although the odontogenic aetiology of the infection would remain complex to demonstrate [44]. Experiences with pre-operative dental screening in orthopaedic surgery highlighted a significant number of older individuals requiring dental intervention [45]. It is not known how much time must elapse between dental and orthopaedic procedures to avoid the risk of the transient bacteraemia-derived PJI. It must be noted, however, that even in the case of an indication for a dental visit before orthopaedic surgery, an older individual could have various impediments to access: an inability to arrange the visit (e.g., distance, homebound), a lack of money or willingness to pay, or odontophobia [46]. There is also a recognised lack of professionals trained in the care of geriatric patients [47]. The notable gap regarding preventive or corrective treatments before surgery has led the current practice to predominantly focus on a few post-operative actions, such as the use of antibiotic prophylaxis during the initial healing period [48]. However, there is room for more comprehensive initiatives. Even in the absence of physical and mental impediments, a patient may eat only half of the calories served in the first days after spine surgery [49], with the other half possibly posing a risk of silent aspiration from unnoticed dental occlusion disorders, xerostomia, periodontal disease, or ill-fitting dentures [50]. Since osteoarthritis can also affect the hands, elbows, and shoulders, it is reasonable to think that even if the patient is able to move to the bathroom and brush their teeth, some may have difficulty handling a toothbrush. Temporomandibular joint deterioration, although subclinical, may also be quite common among older adults, further disrupting related musculature [51]. These and other reasons, which encompass the infectious and nutritional risks linked to an unhealthy mouth, diseases, and/or medications that worsen the symptoms and possible illiteracy about or an inability to apply effective oral hygiene techniques, together with frequent downtimes during hospitalisation, push us to construct a path of quality improvement.

## 6. The Integrated Oral Health Network Applied to Orthopaedic Surgery (IOHN-OS)

We think that an indication for elective orthopaedic surgery is the ideal time to check on oral health and to start what can be defined as the Integrated Oral Health Network applied to Orthopaedic Surgery (IOHN-OS). The scope of a generalist dental consultation is to preserve or restore the desired degree of functionality of the oral cavity. The IOHN-OS would aim to (1) mitigate the patient’s pre-operative vulnerabilities, (2) integrate bedside gerodontology for in-hospital oral hygiene, meal, and eating support, (3) provide educational materials on geriatric dentistry, and (4), if needed, schedule follow-up dental visits. In Figure 1, we report an infographic that illustrates the various phases of the process.

In brief, trained case-manager nurses (or a dental professional if available) could lead the initial oral screening at pre-admission visits using standardised forms specifically developed to discover if there is any oral problem. If so, multidisciplinary management based on the patient’s prevailing clinical and surgical needs will be activated, including a specialist examination when required. Patients with special oral needs could benefit from in-hospital advanced diagnostic tools, such as radiographic equipment and laboratories, which can potentially shorten the time for an accurate and definitive diagnosis. The practice of oral hygiene at the bedside by dental hygienists would enhance the quality of health care and patient experiences and enrich the collaborative peri-operative team of dietitians (e.g., medical nutrition therapy), physiotherapists (e.g., masticatory muscle exercises [52]), and speech pathologists (e.g., chewing and swallowing rehabilitation). Inter-profession and patient interactions would be coordinated by a case-manager nurse and tailored to the post-operative patient’s level of autonomy, mobility, collaboration, and rehabilitative needs. Importantly, the oral consequences of degenerative conditions or medications must be projected over time, anticipating the involvement of other specialists or remodelling pharmacotherapy that poses a threat to oral health. A risk control initiative for potential infectious foci or aspiration pneumonia must be prioritized from a stratified risk management perspective. The continuity of care will be guaranteed even after discharge, with territorial care pathways integrated with the same multidisciplinary and multimodal competence. This would require rebuilding the existing pathways and creating economic, legal, and political conditions that adhere to a geriatric oral health care transition model as a basis for a population-relevant approach [53].

### Limitations

The theoretical IOHN-OS must be interpreted with caution. First, communication between the dental team, the orthopaedic surgeon, and the patient is essential, and without it, this approach is unsuitable. Patients should be activated by the orthopaedic surgeon or any other deputy in charge, such as the case-manager nurse, who takes the initiative. Simply increasing pre-operative awareness about proper oral hygiene is not sufficient to push patients to undergo a dental consultation before orthopaedic surgery [54]. Second, the treatment of some oral problems, such as periodontal diseases and ill-fitting dentures, takes time. The IOHN-OS team, together with the patient, will have to balance the risks/benefits of a temporary pre-operative risk mitigation treatment without postponing orthopaedic surgery against a time-consuming restoration of the desired degree of function of the mouth. Third, the success of the IOHN-OS may strongly depend on where the pre-operative dental consultation is organised. Reduced feasibility could occur when dental visits do not take place where a patient is scheduled to be operated on. A similar inapplicability may also arise when there are no dentists or dental hygienists on-site for bedside gerodontology activities. Fourth, educational initiatives before and after surgery may not be viable for older patients who are routinely depending on a caregiver since it ought to be that the latter are involved in the training programme. Digital technologies for oral health literacy, training, detection, and surveillance like mOralHealth [55] might help. Fifth, specialist oral care, diagnostic, and bedside visits with portable equipment pose an economic obstacle that cannot always be overcome [45]. A similar costly burden may arise when a transversal dental referral with oriented therapy before surgery is imposed by the hospital [41]. The IOHN-OS will need to be more sustainable [43], perhaps providing different degrees of support through the allocation of more comprehensive and in-depth resources to more at-risk and/or vulnerable patients (i.e., health equity). This could also help overcome eventual dental system decay and dentist shortages. In time, investments ought to be made to cover in-hospital dental visits, accompanying diagnostic–therapeutic expenses, and external consultations when an orthopaedic clinic lacks a dental department. Forecasting/longitudinal studies will have to justify the cost.

## 7. Conclusions

As healthcare providers in orthopaedic surgery, we diligently endeavour to correct modifiable pre-operative patients’ vulnerabilities and enhance peri-operative experiences to mitigate the risk of complications and support patients in living a burden-free, happy, and long life after surgery. The theorised IOHN-OS incorporates oral health assessment and management before surgery, in-hospital oral hygiene and support, and both usual and innovative education programmes before and after surgery (e.g., techniques to minimise recessions, fluoride regimens, tailored antibiotic treatment that preserves “good” bacteria [1]). This integrated network has the potential to prolong an older person’s desire to smile and their ability to eat and speak over time. It can also motivate stakeholders to align with the functional, psychosocial, and economic perspectives of patients. Population-based actions like the IOHN-OS may be equally important in dealing with increasing life expectancy, phenotypic complexities, and healthcare demands and costs. It could be the missing piece of a truly holistic, patient-centred approach required in today’s medicine.

## Figures and Tables

**Figure 1 geriatrics-09-00039-f001:**
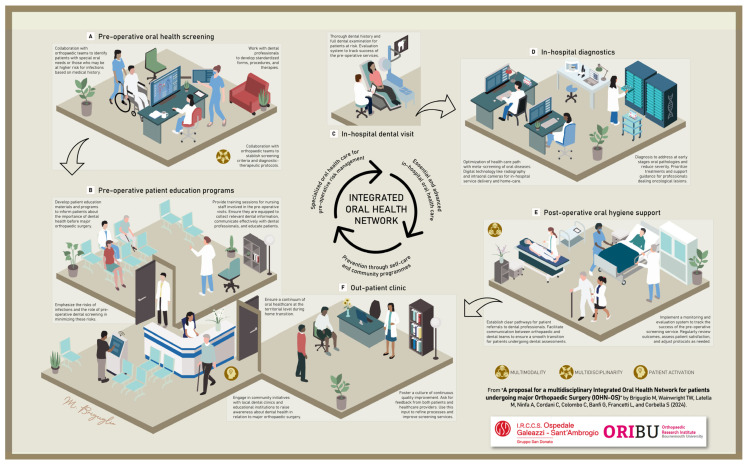
Healthcare network and stages to improve the quality of orthopaedic care with the integration of a dental team. A framework that presents the Integrated Oral Health Network applied to Orthopaedic Surgery (IOHN-OS), designed to enhance the perioperative care trajectory of patients’ vulnerabilities, especially if they are old and malnourished. The successful application of the IOHN-OS would effectively implement an oral risk screening (**A**), diagnosis when needed (**C**,**D**), and corrective initiative before surgery (**B**), further delivering in-hospital services (**E**) and long-term oral health programmes (**F**) for the orthopaedic population.

## Data Availability

Not applicable.

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
