# Peer review of "A Proposal for a Multidisciplinary Integrated Oral Health Network for Patients Undergoing Major Orthopaedic Surgery (IOHN-OS)"

_geriatrics, 2024, doi:10.3390/geriatrics9020039_

Round 1

Reviewer 1 Report

Comments and Suggestions for Authors

The text could be improved by focusing on proposals for dental improvements. As possible limitations, it is indicated that the absence of a dental hygienist could make assistance impossible, which is not true, given that a hygienist works through an exercise of subordination to the dentist, not at the same level, therefore, there may be another assistant or even another dentist who will carry out the work.

Author Response

Dear Reviewer 1,

thank you for your interest in reviewing the manuscript. We have proceeded to implement the text according to your suggestions, which were also proposed by other Reviewers. All the changes made to the text are highlighted in green. In particular, we have included the reference to the dentist for the bed-side gerodontology - section 6.1 Limitations – line 225.

Thank you again for taking the time to read out. We renew our wishes that the new corrected version will satisfy the Editorial Office and thus make the manuscript worthy of publication.

Best regards
The authors

Reviewer 2 Report

Comments and Suggestions for Authors

Please, see attached file for details.

Author Response

Dear Reviewer 2,

thank you for your interest in reviewing the manuscript. We have proceeded to implement the text according to your suggestions. All the changes made to the text are highlighted in green. We hope that the new corrected version will satisfy you and the Editorial Office and thus make the manuscript worthy of publication.

Best regards
The authors

  1. Perhaps you could choose a title that represents the content of the manuscript a little better. E.g.: Need for a multidisciplinary Integrated Oral Health Network for patients with major orthopaedic surgery.

We changed the title in “A proposal for a multidisciplinary Integrated Oral Health Network for patients undergoing major Orthopaedic Surgery (IOHN-OS)”.

  1. Please add "senior dentistry", "care management", and "transition" to the keywords.

The maximum number of keywords must be ten. We prefer to include only MeSH terms among the keywords and “senior dentistry” is not a MeSH term. We substituted “Continuity of Patient Care” with “Hospital to Home Transition”. We substituted “Deglutition Disorders” with “Patient Care Management”.

  1. Section 2 «Aging poses a risk to oral cavity of older individuals“: Please, add references. They are missing between lines 64-71. This also applies to the last sentence of this paragraph.

Thanks for the comment. The authors would like to point out to the Reviewer that the opinion article reflects the point of view of specialists in the field. Therefore, we have not always considered it necessary to provide references to support our declarations, sometimes born from experience and at times narrating physiology. However, we agree on adding reference on some relevant points of the paragraph between line 61 and 81.

  1. The sentence «Edentulism and ill-fitting dental implants are associated with reduced masticatory efficiency.“:
    1. Please make sure that you have correctly distinguished between bite force and chewing efficiency throughout the article. These are different parameters, which also have different influencing factors. You can find more information on this here: Jockusch J, Hopfenmüller W, Nitschke I. Chewing function and related parameters as a function of the degree of dementia: Is there a link between the brain and the mouth? J Oral Rehabil. 2021 Oct;48(10):1160-1172. doi: 10.1111/joor.13231. Epub 2021 Aug 14. PMID: 34288029; PMCID: PMC9291087.
    2. What do you mean by "ill-fitting implants"? This is not a common term in dentistry. Implants are artificial tooth roots, so I'm not assuming you mean this but dentures, which are generally removable and can be ill-fitting if not fitted and regularly followed up by a dentist. This in turn can negatively affect chewing efficiency but also bite force. Please clarify this again and look for literature that supports these statements on bite force and chewing efficiency.
    3. There is also no bibliography for this sentence. Please adapt or replace with the predicted one. Thank you.

We are aware of the diversity of terms bite force ≠ chew efficiency. Thank you for sharing the reference. Thank you also for pointing out the importance of highlighting the difference to the reader. We corrected the term and added the reference on line 79-80.

  1. Please change the sentence (line 78) to: Impact of medications and conditions on oral health of older patients

We substituted “conditions” with “illnesses”.

  1. Bitte stellen Sie bei diesem Satz klar, um welche Mineralien es sich handelt und geben Sie eine Literaturstelle an (Zeile 89-91: Similar consequences on the homeostasis of minerals that are essential for dental health are elicited by loop diuretics, which are frequently used in disorders of sodium balance.).

We clarified and integrated the sentence on line 93-95.

  1. Please change osteonecrosis of the jaw (line 92/93) to BRONJ (Bisphosphonate related osteonecrosis of the jaw (BRONJ)) in this sentence as you are referring to bisphonates. Also add a reference for this. Please also clarify that these bisphosphonates are not only used for the treatment of osteoporosis.

We do not think that it is necessary to say that bisphosphonates are used only for treating osteoporosis. Moreover, we wanted to include all types of causes of osteonecrosis of the jaw, of whom bisphosphonates use is only one example (others are from antineoplastics, radiation-induced or infection). Therefore, we didn’t integrate the text with the acronym BRONJ but removed the indication of osteoporosis (line 99)

  1. Line 99/100: „Oral health problems not only affect what happens from the neck up, but they can also indirectly and negatively influence eating behaviours.“ Please avoid using colloquial language. Please rephrase and find sources for the consequences in the area of nutrition. It is not primarily the eating behavior that is influenced, but the inability to chew (bite force), to grind food (chewing efficiency) and possibly also to swallow.

We corrected and integrated the text with references as suggested (line 101-103).

  1. The same applies to the following sentence: The provision of dentures is not necessarily the reason why the chewing process is impaired. Please also rewrite this thought (line 1001/102). Please do not use the word "dental appliances" in this context either, you probably mean removable dentures, 2 right? If it really is "dental appliances", then please provide a source for this statement. All in all, the literature citation is also missing here.

Yes, we meant dentures. And no, the presence of dentures is not necessarily the reason why chewing is impaired. But it may be one of the reasons. We extensively corrected the text the be more evidence-based and understandable by the reader (line 103-109).

  1. Line 105/106, please add literature. («L-ascorbic acid deficiency causes periodon-105 tal and gum diseases and bleeding on probing»)

We added a reference on line 119.

  1. Line 134, please add literatur («It is commonly believed that infection foci in the oral cavity at the time of major 133 orthopaedic surgery may increase the risk of periprosthetic joint infection (PJI).»)

We added a reference and slightly changed the phrase on line 144-145.

  1. Line 140, please add literature («It is not known how much time must elapse between the dental and orthopaedic pro-139 cedures to avoid the risk of the transient bacteremia-derived PJI.»)

We do not think a reference should be added here. This is what the authors support.

  1. It is unusual that physiotherapist do masticatory muscle excersises. Normally the try to reduce the muscle tension. Therefore, maybe you would like to cite this publication to give a hint to physiotherapists: Jockusch J, Hahnel S, Sobotta BBAJ, Nitschke I. The Effect of a Masticatory Muscle Training Program on Chewing Efficiency and Bite Force in People with Dementia. Int J Environ Res Public Health. 2022 Mar 22;19(7):3778. doi: 10.3390/ijerph19073778. PMID: 35409460; PMCID: PMC8997984.

Thanks for the sharing. The authors believe and support masticatory muscle exercise. Of more usage and training and the more the masticatory muscle thickness increases. We added the reference you suggested on line 192.

  1. In the field of senior dentistry, models are also being discussed as to how a patient's transition from one phase of life to another can be successfully mastered and dental care maintained. For the sake of completeness, could you include and name these concepts? The objectives pursued here are similar to those in your model. You can find more information here: Nitschke I, Nitschke S, Haffner C, Sobotta BAJ, Jockusch J. On the Necessity of a Geriatric Oral Health Care Transition Model: Towards an Inclusive and Resource-Oriented Transition Process. Int J Environ Res Public Health. 2022 May 18;19(10):6148. doi: 10.3390/ijerph19106148. PMID: 35627684; PMCID: PMC9141301.

We thank you for the sharing. We integrated the text with a reference to the geriatric oral health care transition model on line 201-204.

  1. Please discuss possible problems with the implementation of the concept in other countries in the Limitations section. For example, with payment in Germany / Switzerland, a dentist must be paid by the hospital if he treats patients in a hospital who are treated there by doctors as inpatients. This means that dentists are often not called in or are unable to treat patients. You also mention that treatments should be carried out on site. Please also discuss the fact that this also requires the necessary infrastructure on the part of the dentists (mobile dentistry) and/or the hospital. Not all treatments can be carried out at the patient's bedside. Equipment, materials or diagnostic tools, such as dental x-rays, may be required.

We agree with the viewpoint. We integrated the limitation section in the discussion with your notes on line 236-239.

  1. Line 217: What does that mean? »or fight fire with fire“

The expression refers to the suggested possibility that future PJI prevention might rely on the promotion of “good” bacteria to fight “bad” bacteria. We rephrased on line 247.

  1. Please check the reference list: For expample #17 the first author is not correct. Anne Marie ist he first name and not the surname.

Thanks for the comment. We adhered to the citation from PubMed, and in PubMed it is cited as Anne Marie U et al. It is not possible to change the citation so as not to lose the reference. The doi will be there to guarantee the connection.

Reviewer 3 Report

Comments and Suggestions for Authors

The manuscript entitled “Is the old patient in the hospital for major orthopaedic surgery?  We should check on oral health conditions” is interesting and carries a very important message and the possibility of implementation in the future. However, there are some issues that needs to be addressed as part of the major revision.

Page 2, lines 65/69 and 80-81 where xerostomia is introduces as the part of oral health problems, references are missing. As xerostomia is rarely only age-related but caused by medications, it is necessary to include it in the discussion as one of the additional problems to be solved, because it requires the involvement of other specialists about these medications prescriptions and possibility of replacement.

The term "ill-fitting dental implants", page 2, line 75 is not appropriate. Dentures can be ill fitting, not implants.

In section 4, the consequences of general poor health on the oral cavity and orofacial system (such as the consequences of anemia on the oral mucosa) and the consequences of poor oral health on food choices with potential resulting malnutrition intertwine. Although both directions of influence are important, it is necessary to separate them and indicate the more important direction for this topic, which is how oral health affects food choices and malnutrition. If the idea is to recognize already malnourished patients and examine oral health in more detail, then explain it differently.

​References useful for this section are: Sheiham, A.; Steele, J. Does the condition of the mouth and teeth affect the ability to eat certain foods, nutrient and dietary intake and nutritional status amongst older people? Public Health Nutr. 20014, 797–803. Pedersen, A.M.; Bardow, A.; Jensen, S.B.; Nauntofte, B. Saliva and gastrointestinal functions of taste, mastication, swallowing and digestion. Oral Dis. 20028, 117–129.)

Given that section 5 lists many reasons for poor oral health, such as periodontal disease and edentulism, it is necessary to highlight in the discussion how these are the main problems for the implementation of this path ((IOHN-OS), because the treatment of periodontal disease as well as denture production, especially in edentulous patients, requires time and gives partial success in older adults with poor general health. Bearing this in mind, it would be beneficial to focus on feasible options in this pathway, which is to solve the oral infection with shorter interventions (extractions and causal periodontal disease treatment) that will prevent periprosthetic joint infection and other complications such as aspiration pneumonia. 

Reference 33 does not refer to the lack of dentists who have knowledge of gerodontology (although that statement is true) but to how much knowledge dentists have about elderly patients, which is not equivalent. (Please, try with these references: AE Kossioni. Is Europe prepared to meet the oral health needs of older people?Gerodontology, Volume29, Issue2; June 2012: e1230-e1240. Nilsson A, Young L, Glass B, Lee A. Gerodontology in the dental school curriculum: A scoping review. Gerodontology Volume38, Issue4; December 202: Pages 325-337)

In section 6, when explaining how "The Integrated Oral Health Network" should function, there is a lot of ambiguity as to how the process should be carried out. The nurse in charge of the patient is not able to assess the patient's need for dental treatment. Although it is said in the limitations that there is a potential problem in the communication between the dentist and the surgical team, it remains unclear who determines what could be a potential problem in oral health and a risk for surgical recovery and what treatment should be done. The remark that the hospital is generally equipped with more advanced diagnostic tools than dental practices in the community is somehow confusing because there is no emphasis on those patients with suspected malignancies. Furthermore, it is not clear how and where dental treatment should carried out, stating only about the importance of the oral hygienist involvement, which is indisputable, but it is certainly only an addition after the dental treatment has been carried out.

Author Response

Dear Reviewer 3,

thank you for your interest in reviewing the manuscript. We have proceeded to implement the text according to your suggestions. All the changes made to the text are highlighted in green. We hope that the new corrected version will satisfy you and the Editorial Office and thus make the manuscript worthy of publication.

Best regards
The authors

  1. Page 2, lines 65/69 and 80-81 where xerostomia is introduces as the part of oral health problems, references are missing. As xerostomia is rarely only age-related but caused by medications, it is necessary to include it in the discussion as one of the additional problems to be solved, because it requires the involvement of other specialists about these medications prescriptions and possibility of replacement.

Thanks for the comment. We integrated some references in section 2, line 68 and on. Also, we added the concept you suggested in the discussion line 195-198.

  1. The term "ill-fitting dental implants", page 2, line 75 is not appropriate. Dentures can be ill fitting, not implants.

We corrected.

  1. In section 4, the consequences of general poor health on the oral cavity and orofacial system (such as the consequences of anemia on the oral mucosa) and the consequences of poor oral health on food choices with potential resulting malnutrition intertwine. Although both directions of influence are important, it is necessary to separate them and indicate the more important direction for this topic, which is how oral health affects food choices and malnutrition. If the idea is to recognize already malnourished patients and examine oral health in more detail, then explain it differently.

Thanks for the comment. We agree with the idea that the two factors are related. We also agree with the comment that the paragraph should start describing how oral problems affects nutrition. We rewrote the paragraph between line 101 and 124 accordingly and added some new references.

  1. References useful for this section are: Sheiham, A.; Steele, J. Does the condition of the mouth and teeth affect the ability to eat certain foods, nutrient and dietary intake and nutritional status amongst older people? Public Health Nutr. 2001, 4, 797–803. Pedersen, A.M.; Bardow, A.; Jensen, S.B.; Nauntofte, B. Saliva and gastrointestinal functions of taste, mastication, swallowing and digestion. Oral Dis. 2002, 8, 117–129.)

We added the suggested reference (line 105).

  1. Given that section 5 lists many reasons for poor oral health, such as periodontal disease and edentulism, it is necessary to highlight in the discussion how these are the main problems for the implementation of this path ((IOHN-OS), because the treatment of periodontal disease as well as denture production, especially in edentulous patients, requires time and gives partial success in older adults with poor general health. Bearing this in mind, it would be beneficial to focus on feasible options in this pathway, which is to solve the oral infection with shorter interventions (extractions and causal periodontal disease treatment) that will prevent periprosthetic joint infection and other complications such as aspiration pneumonia.

Thanks for the valuable comment. We agree with you in presenting a feasible approach to the reader. We integrated in the discussion line 198-199.

  1. Reference 33 does not refer to the lack of dentists who have knowledge of gerodontology (although that statement is true) but to how much knowledge dentists have about elderly patients, which is not equivalent. (Please, try with these references: AE Kossioni. Is Europe prepared to meet the oral health needs of older people?Gerodontology, Volume29, Issue2; June 2012: e1230-e1240. Nilsson A, Young L, Glass B, Lee A. Gerodontology in the dental school curriculum: A scoping review. Gerodontology Volume38, Issue4; December 202: Pages 325-337)

Thanks for the correction. We changed the reference and slightly modified the text line 154-155.

  1. In section 6, when explaining how "The Integrated Oral Health Network" should function, there is a lot of ambiguity as to how the process should be carried out. The nurse in charge of the patient is not able to assess the patient's need for dental treatment. Although it is said in the limitations that there is a potential problem in the communication between the dentist and the surgical team, it remains unclear who determines what could be a potential problem in oral health and a risk for surgical recovery and what treatment should be done. The remark that the hospital is generally equipped with more advanced diagnostic tools than dental practices in the community is somehow confusing because there is no emphasis on those patients with suspected malignancies. Furthermore, it is not clear how and where dental treatment should carried out, stating only about the importance of the oral hygienist involvement, which is indisputable, but it is certainly only an addition after the dental treatment has been carried out.

We agree. We are not sure there will be a one-size fits all solution pathway and hence we do not suggest one-for-all. Key point here is that we want to make the reader start thinking about that there needs to be a pathway. The mechanics of how it is delivered will depend on context and funding and will therefore be localised.

For instance, we partially agree that the nurse in charge of the patient is not able to assess the patient’s need for dental treatment. But it must be remembered that our proposal applies in a context where the presence of the dentist or dental hygienist at pre-admission is new. It is not feasible to think that at the orthopedic pre-admission, together with the anaesthetist, surgeon and nurse, there is always also a dentist or dental hygienist. Otherwise we would also have to include a dietician, physiotherapist, speech therapist, and other healthcare professionals. Considering that in large orthopaedic centres, such as in our hospital, there are dozens of pre-admissions a day.

The authors think that, at least for the moment with the current scarcity of resources, it is more feasible for the nurse -appropriately trained- to carry out the initial screening of patients’ mouths. In case of doubt or positivity, there will be a referral to the specialist. As regards advanced diagnostics, we refer to the presence of advanced diagnostic tools in hospital. Something that is difficult to do in some dental practices in the community. As for where to get treatment, there is no one-size-fits-all pathway. Certainly, a patient stratification system will have to be created to ensure that the management of the most serious cases can be deferred to the dentist. We clarified the text according to your comment in section 6 and 6.1.

Round 2

Reviewer 2 Report

Comments and Suggestions for Authors

Dear authors, 

thank you for revising the manuscript according to the proposed aspects.

Still the reference number 33 is incorrect cited. The first authors name is wrong: First Name is cited as Surname. Please change.

Thank you, best regards

Reviewer 3 Report

Comments and Suggestions for Authors

I am satisfied with your responses to my comments and with your edits in the text accordingly. Also, the additional explanation in the answer gives me confirmation that we agree on the application of this concept, its further development and importance.